# MRI-Based Radiomics Combined with Deep Learning for Distinguishing IDH-Mutant WHO Grade 4 Astrocytomas from IDH-Wild-Type Glioblastomas

**DOI:** 10.3390/cancers15030951

**Published:** 2023-02-02

**Authors:** Seyyed Ali Hosseini, Elahe Hosseini, Ghasem Hajianfar, Isaac Shiri, Stijn Servaes, Pedro Rosa-Neto, Laiz Godoy, MacLean P. Nasrallah, Donald M. O’Rourke, Suyash Mohan, Sanjeev Chawla

**Affiliations:** 1Translational Neuroimaging Laboratory, The McGill University Research Centre for Studies in Aging, Douglas Hospital, McGill University, Montréal, QC H4H 1R3, Canada; 2Department of Neurology and Neurosurgery, Faculty of Medicine, McGill University, Montréal, QC H3A 2B4, Canada; 3Department of Electrical and Computer Engineering, Kharazmi University, Tehran 15719-14911, Iran; 4Rajaie Cardiovascular Medical and Research Center, Iran University of Medical Science, Tehran 19956-14331, Iran; 5Division of Nuclear Medicine and Molecular Imaging, Geneva University Hospital, CH-1211 Geneva, Switzerland; 6Department of Radiology, Perelman School of Medicine at the University of Pennsylvania, Philadelphia, PA 19104, USA; 7Department of Pathology and Laboratory Medicine, Perelman School of Medicine, University of Pennsylvania, Philadelphia, PA 19104, USA; 8Department of Neurosurgery, Perelman School of Medicine, University of Pennsylvania, Philadelphia, PA 19104, USA

**Keywords:** grade 4 astrocytoma, glioblastoma, isocitrate dehydrogenase mutation, conventional magnetic resonance imaging, radiomics, machine learning, deep learning

## Abstract

**Simple Summary:**

To differentiate IDH-mutant grade 4 astrocytomas from IDH-wild-type glioblastomas, two MRI sequences (post-contrast T1 and T2-FLAIR) were acquired from 57 patients. The images were resliced, resampled, and realigned. In the next step, tumors were segmented semi-automatically into subregions including whole tumor, edema region, core tumor, enhancing region, and necrotic region. A total of 105 radiomic features were extracted from each subregion. The data were divided randomly into training and testing sets. A deep learning-based data augmentation method (CTGAN) was implemented to synthesize 200 datasets. A total of 18 classifiers were used to distinguish two genotypes of grade 4 astrocytomas. The best discriminatory power was obtained from core tumor regions overlaid on post-contrast T1 using the K-best feature selection algorithm and a Gaussian naïve Bayes classifier.

**Abstract:**

This study aimed to investigate the potential of quantitative radiomic data extracted from conventional MR images in discriminating IDH-mutant grade 4 astrocytomas from IDH-wild-type glioblastomas (GBMs). A cohort of 57 treatment-naïve patients with IDH-mutant grade 4 astrocytomas (*n* = 23) and IDH-wild-type GBMs (*n* = 34) underwent anatomical imaging on a 3T MR system with standard parameters. Post-contrast T1-weighted and T2-FLAIR images were co-registered. A semi-automatic segmentation approach was used to generate regions of interest (ROIs) from different tissue components of neoplasms. A total of 1050 radiomic features were extracted from each image. The data were split randomly into training and testing sets. A deep learning-based data augmentation method (CTGAN) was implemented to synthesize 200 datasets from the training sets. A total of 18 classifiers were used to distinguish two genotypes of grade 4 astrocytomas. From generated data using 80% training set, the best discriminatory power was obtained from core tumor regions overlaid on post-contrast T1 using the K-best feature selection algorithm and a Gaussian naïve Bayes classifier (AUC = 0.93, accuracy = 0.92, sensitivity = 1, specificity = 0.86, PR_AUC = 0.92). Similarly, high diagnostic performances were obtained from original and generated data using 50% and 30% training sets. Our findings suggest that conventional MR imaging-based radiomic features combined with machine/deep learning methods may be valuable in discriminating IDH-mutant grade 4 astrocytomas from IDH-wild-type GBMs.

## 1. Introduction

Glioblastomas (GBMs) are devastating and universally fatal brain cancers in adults despite advancements in diagnostic and therapeutic strategies [1]. Approximately 14,000 new cases of GBM are diagnosed in the US each year, with an estimated incidence of 3.19 per 100,000 people [2]. In recent years, the emergence of molecular profiling in neuro-oncology has had a considerable bearing on the classification, diagnosis, prognosis, and clinical management of GBM patients [3]. The 2016 WHO classification system recognized the somatic mutation of the isocitrate dehydrogenase (IDH) gene in gliomas as a distinct entity regardless of histopathological features [4]. IDH mutation occurs in 50–70% of WHO grade 2/3 gliomas and 10% of GBMs [5], which has been considered as a new paradigm in determining the prognosis of these patients. The new 2021 WHO system has reclassified GBMs as IDH-mutant grade 4 astrocytomas or IDH-wild-type GBMs based on gene expression profiles [6]. It has been widely reported that glioma patients harboring IDH mutations demonstrate a better response to chemoradiation therapy and live longer than those with IDH-wild-type alleles [7,8]. Immunohistochemical analyses and exomic sequencing are considered the gold standard for determining IDH mutation status in gliomas [9,10]; however, several factors, such as tissue heterogeneity, partial sampling of tissue specimens, and presence of variable amounts of antigens constrain the utility of these methods in reliable detection of IDH mutation status [11]. Moreover, it is not always possible to perform neurosurgical interventions because of the eloquent locations of these neoplasms.

Therefore, non-invasive identification of IDH-mutant gliomas is vital for making informed decisions on therapeutic intervention and prognosticating these patients. IDH mutations confer the neomorphic activity of an enzyme leading to the conversion of alpha-ketoglutarate (α-KG) to 2-hydroxyglutarate (2HG) [12]. Prior studies [13,14,15] have reported the clinical utility of modified MR spectroscopy sequences in identifying IDH-mutant gliomas by detecting characteristic resonances of 2HG. However, not all IDH-mutant gliomas show the neomorphic activity of the 2-HG production [16]. Moreover, these sophisticated spectroscopic sequences are not readily available in routine clinical settings.

Conventional magnetic resonance imaging (MRI) remains the mainstay for determining tumor location, size, and structural features in neurooncology [17]. Radiomics is a rapidly evolving translational field that automatically produces mineable high-dimensionality data from positron emission tomography (PET) [18,19], computed tomography (CT), and MRI images with high precision [20,21,22]. Several previous studies have documented the clinical potential of quantitative radiomic features extracted from conventional MRI data in diagnosis, determining molecular signatures, assessing treatment response, and predicting survival outcomes in GBM patients [23,24,25,26,27,28]. Some other studies have also reported promising findings in identifying IDH-mutant grade 4 astrocytomas using conventional neuroimaging-based radiomic classification models with variable accuracies [29,30]. However, these studies were limited by the extraction of a sparse number of radiomic features (*n* = 31) [29] or by the inclusion of a small sample size of IDH-mutant grade 4 astrocytomas (*n* = 7) [30].

With these inadequacies in mind, the current study was designed to investigate the potential utility of radiomic features extracted from different tumor habitats as visible on widely available and universally acquired preoperative post-contrast T1 weighted and T2-FLAIR images in differentiating IDH-mutant grade 4 astrocytomas from IDH-wild-type GBMs.

## 2. Materials and Methods

### 2.1. Patient Population

This retrospective study was approved by the institutional review board and was compliant with the Health Insurance Portability and Accountability Act. The inclusion criteria for enrollment in the present study were that all patients had (a) histopathologically confirmed grade 4 astrocytoma according to the WHO classification system, (b) a known IDH mutation genotype using immunohistochemistry and/or gene sequencing, and (c) available preoperative anatomical MR images acquired using identical data acquisition protocol. Based upon the inclusion criteria, a cohort of 57 patients (mean age = 57.7 ± 6.9 years, 39 males and 18 females) with newly diagnosed grade 4 astrocytoma and GBM were recruited in this study. Of these 57 patients, 23 had the IDH-mutant genotype, and 34 had the IDH-wild-type genotype.

### 2.2. Determination of IDH Mutational Status by Immunohistochemistry and Sequencing

Hematoxylin, eosin staining, and immunohistochemistry were conducted on 5-micron thick, formalin-fixed (10%), paraffin-embedded tissue sections mounted on Leica Surgipath slides followed by drying for 60 min at 70 °C. In addition, immunohistochemistry to detect the IDH1 p.R132H variant was performed by using an anti-IDH1-R132H antibody (monoclonal mouse anti-human IDH1 (R132H), Dianova, DIA Clone H09) and DAB chromogen was performed on a Leica Bond III instrument using a bond polymer refine detection system (Leica Microsystems AR9800) following a 20-min heat-induced epitope retrieval with Epitope Retrieval 2, EDTA, pH 9.0. Appropriate positive and negative controls were included.

In addition, massively parallel sequencing or RealTime polymerase chain reaction (PCR) was performed to confirm the immunohistochemical results and to interrogate other IDH variants. For RealTime PCR, formalin-fixed, paraffin-embedded (FFPE) specimens with >20% tumor content were analyzed for IDH1 and IDH2 variants using Abbott RealTime Assays (Abbott Molecular, Inc., Abbott Park, IL, USA) after extraction using the QIAamp DSP DNA FFPE Tissue Kit (Qiagen, Hilden, Germany). The Abbott RealTime IDH1 assay detects 5 single nucleotide variants (SNVs) in IDH1 (p.R132C, p.R132H, p.R132G, p.R132S, and p.R132L). The Abbott RealTime IDH2 assay detects 9 SNVs in IDH2 (p.R140Q, p.R140L, p.R140G, p.R140W, p.R172K, p.R172M, p.R172G, p.R172S, and p.R172W). The Abbott m2000rt software performs variant calling, and results are qualitatively reported as positive or not detected. Tests were performed according to the manufacturer’s instructions by adding a dilution step to the IDH2 assay. For massively parallel sequencing, the panel gives full gene coverage of 152 genes, using the Agilent Haloplex design with unique molecular identifiers as described previously [31]. Briefly, DNA was extracted from FFPE or specimens preserved in PreservCyt. Samples were multiplexed and sequenced on a HiSeq with total deduplicated reads of 6.5 million/sample; duplicate reads were removed based on incorporating unique molecular identifiers. All variants were identified using an in-house data processing bioinformatics pipeline capable of detecting SNVs, insertions and/or deletions (indels), and copy number gains for a subset of genes based on increased read depth. An experienced neuropathologist (MPN) reviewed cases from all patients to confirm the IDH status.

### 2.3. MRI Data Acquisition

All patients underwent an MRI on a 3T Tim Trio whole-body MR scanner (Siemens, Erlangen, Germany) equipped with a 12-channel phased array head coil. The anatomical imaging protocol included an axial 3D-T1-weighted magnetization-prepared rapid acquisition of gradient echo (MPRAGE) imaging [repetition time (TR)/echo time (TE)/inversion time (TI) = 1760/3.1/950 ms]; in-plane resolution = 1 × 1 mm^2^; slice thickness = 1 mm; the number of slices = 192; and axial T2-FLAIR imaging (TR/TE/TI = 9420/141/2500 ms, slice thickness = 3 mm; the number of slices = 60). The post-contrast T1-weighted images were acquired with the same parameters as the pre-contrast acquisition after administration of the standard dose of gadobenate dimeglumine (MultiHance, Bracco Imaging, Milano, Italy) intravenous contrast agent using a power injector (Medrad, Idianola, PA, USA).

### 2.4. Image Processing

The overview of the image processing and radiomics pipeline, which includes image registration, tissue segmentation, feature extraction, feature selection, and radiomics model building, is shown in Figure 1. An investigator (SAH) blinded to the IDH mutational status performed all image processing steps. Post-contrast T1-weighted images were resliced, resampled, and co-registered with T2-FLAIR images using a linear affine transformation. A semi-automatic segmentation approach was used to generate regions of interest (ROIs) on the anatomical images. Care was taken to exclude surrounding normal brain vessels. Manual inspections were performed by an experienced neuroradiologist to correct for any pixel anomalies present within the ROIs. Accordingly, these ROIs were modified manually by adding pixels for tumor regions not included in the initial ROIs or by removing pixels for non-tumor regions included in the initial ROIs. Post-contrast T1 weighted images were used to segment solid/contrast-enhancing regions, necrotic regions, and core tumors (solid + necrotic region). T2-FLAIR images were used to segment peri-tumoral edematous regions and whole tumor volumes. All tissue segmentations were performed using 3D slicer software. To maximize the characterization of tumors, these 5 segmented ROIs were overlaid on the source post-contrast T1-weighted images and T2-FLAIR images for the data analysis (Figure 2 and Figure 3). A bias field correction using N4 and an image normalization using histogram matching were performed using the 3D slicer software on the MRI images before feature extraction to avoid any potential bias field distortions and data heterogeneity bias.

### 2.5. Radiomic Feature Extraction

From each segmented ROI, 105 original radiomic features from categories (shape, first-order statistical, second-order texture, and higher-order statistic) were extracted using the PyRadiomics package in python [32]. These original features can be sub-divided into 7 classes, including 13 shape features, 18 first-order statistical features, 23 gray level co-occurrence matrix (GLCM) features, 14 gray level dependence matrix (GLDM) features, 16 gray level size zone matrix (GLSZM) features, 16 gray level run length matrix (GLRLM) features, and 5 neighboring gray-tone difference matrix (NGTDM) features. Altogether, 525 radiomic features were extracted from 5 ROIs of each image for a total of 1050 features from post-contrast T1- and T2-FLAIR images. The radiomics features used comply with the standard described by the Imaging Biomarker Standardization Initiative (IBSI) [33]. A high-performance computer system with 16GB RAM and an Intel Core i7-7700 CPU processor @3.60 GHz was used for data processing. The feature extraction took an average of 2–3 min per patient image set. A list of all features is summarized in Appendix A.

### 2.6. Radiomics Feature Selection/Dimension Reduction

It is important to eliminate irrelevant or redundant variables that may cause data overfitting and may bias the performance of the prediction model. Multiple feature selection algorithms, including recursive feature elimination (RFE), minimum redundancy, maximum relevance (mRmR), and K-best were employed to select image features. Patients were divided into 2 mutually exclusive training (80%, 50%, and 30%) and testing (20%, 50%, and 70%) sets using the random shuffling method. Ten percent of the training set was split off to serve as the validation set. All data were normalized by MinMax normalization. The mRmR feature selection technique was used to select 15 features.

### 2.7. Deep Learning Approach for Data Augmentation

The current study implemented a deep learning method based on generative adversarial networks (GAN) for data augmentation [34]. CTGAN is a GAN-based deep learning data synthesizer to increase the number of our datasets that can improve the reproducibility and discriminatory power of radiomics features [35,36,37]. After splitting the data set and selecting bold features using various feature selection algorithms, the selected radiomic features from each model with the highest number were used as the input value for CTGAN to synthesize 200 radiomic features. As a result, after splitting 80%, 50%, and 30% of 57 original data for the training sets, 245, 228, and 217 datasets (80%, 50%, and 30% of 57 + 200 = 245, 228, and 217), including original and generated data, were synthesized, respectively. Different splitting percentages were used to confirm our findings [38] and to prevent the impact of data leakage in our results [39]. Furthermore, a random noise (normal distribution, mean = 0.0, standard deviation = 0.05) [40] was added to the training set. The test sets were not generated, and the original datasets were used for the testing sets.

### 2.8. Machine Learning Classifiers for Prediction Model Building

To develop a prediction model for distinguishing IDH-mutant grade 4 astrocytomas from IDH-wild-type GBMs, a total of 18 single and ensembled machine learning classifiers [Bernoulli naïve Bayes (BNB), multilayer perceptron (MLP), support vector classifier (SVC), Gaussian naïve Bayes (GNB), quadratic discriminant analysis (QDA), bagging classifier, linear discriminant analysis (LDA), logistic regression (RG), ridge, ada boost (AD), hist gradient boosting (HGB), K-neighbors (KN) (K = 5), random forest (RF), gradient boosting (GB), extra trees (ET), decision tree (DT), nearest centroid (NC), and passive aggressive (PA] were employed using an in-house-developed python package. All cases in the training cohort (80%, 50%, and 30%) were used to train the classifiers, and an internal validation (cross-validation) was performed from the testing cohort (20%, 50%, and 70%). Receiver operative characteristic (ROC) curve analyses were performed to evaluate the diagnostic potentials of prediction models in distinguishing 2 groups (IDH-mutant grade 4 astrocytomas and IDH-wild-type GBMs). Area under the ROC curve (AUC), area under the precision-recall curve (PR_AUC), accuracy (ACC), sensitivity, specificity, and negative and positive predictive values (NPV and PPV, respectively) were determined for each prediction model as performance metrics.

## 3. Results

When original MRI data (*n* = 57) were used in discriminating IDH-mutant grade 4 astrocytomas from IDH-wild-type GBMs, the best discriminatory performance (AUC = 0.93, ACC = 0.92, sensitivity = 1, specificity = 0.86, PR_AUC = 0.92) was obtained from solid/contrast enhancing, and core tumor (solid + necrotic region) overlaid on post-contrast T1-weighted images using various combinations of feature selection algorithms and machine learning classifiers. The predictive power, accuracy, sensitivity, specificity, and PR_AUC of the best 10 methods in distinguishing two genotypes of grade 4 astrocytomas are summarized in Table 1.

The relative importance of the best 10 methods in terms of predictive power, accuracy, sensitivity, specificity, and PR_AUC in discriminating two genotypes of grade 4 astrocytomas by using various combinations of feature selection algorithms, machine learning classifiers, and segmented tumor regions when 80%, 50%, and 30% of the generated data were used as training sets are summarized in Table 2, Table 3, and Table 4, respectively. From generated data using 80% as the training set (Table 2), the best discriminatory power (AUC = 0.93, accuracy = 0.92, sensitivity = 1, specificity = 0.86, and PR_AUC = 0.92) in distinguishing two genotypes of grade 4 astrocytomas was obtained from core regions overlaid on post-contrast T1 images when K-best and RFE feature selection algorthims and GNB and PA classifiers were applied. A similar high-diagnostic performance was obtained from enhancing regions overlaid on T2-FLAIR images when the K-best feature selection algorithm and DT and bagging classifiers were applied. From generated data using 50% as the training set (Table 3), necrotic regions of co-registered, post-contrast T1 images with mRmR feature selection and bagging and RF classifiers and the edematous region of the co-registered, post-contrast T1 image with mRmR feature selection and KN classifier provided the highest predictive power (AUC = 0.92, accuracy = 0.92, sensitivity = 0.91, specificity = 0.94, and PR_AUC = 0.93). From generated data using 30% as the training set (Table 4), the core regions of co-registered, post-contrast T1 images with K-best feature selection and LR classifier provided the highest predictive power (AUC = 0.91, accuracy = 0.92, sensitivity = 0.86, specificity = 0.96, and PR_AUC = 0.92).

Heatmaps of predictive power (AUC), predictive accuracy (ACC), sensitivity (SEN), and specificity (SPE) for discriminating IDH-mutant grade 4 astrocytomas from IDH-wild-type GBMs utilizing a variety of feature selections (training set equal to 80%), and machine learning algorithms applied to distinct subregions of neoplasms, are shown in Appendix A, respectively. In addition, the comprehensive findings from using a multi-segmentation approach, feature selection algorithms, and multi-machine learning classifiers in discriminating IDH-mutant grade 4 astrocytomas from IDH-wild-type GBMs in original and generated data with different training and testing sets are provided in the Appendix A.

## 4. Discussion

In this study, we investigated the clinical utility of a conventional neuroimaging-based radiomics approach with deep learning in determining the IDH status of grade 4 astrocytomas. A total of 1050 radiomic features were extracted from different tumor habitats (solid/contrast enhancing, central necrotic, peritumoral edematous, core tumor, and whole tumor regions), encompassing post-contrast T1-weighted and T2-FLAIR images. Our work is an extension of previous studies as we used a GAN-based algorithm to increase our sample size and used a large number of machine learning classifiers (*n* = 18) to build a reliable prediction model in distinguishing IDH-mutant grade 4 astrocytomas and IDH-wild-type GBMs. In the testing cohort, our best prediction model consisted of a central necrotic region from post-contrast, T1-weighted images when a combination of the K-best feature selection algorithm and Gaussian naïve Bayes classifier were used together. This prediction model achieved a high diagnostic performance (AUC = 0.93, accuracy = 0.92, sensitivity = 1, specificity = 0.86, PR_AUC = 0.92) in discriminating two genotypes of grade 4 astrocytomas.

IDH mutation has been recognized as one of the most important molecular markers for diagnosis of gliomas and GBMs based on the 2016 WHO classification system. In addition, according to the recent 2021 WHO classification of tumors of the central nervous system (CNS) [6], previously called IDH-mutant GBM, is now designated as IDH-mutant grade 4 astrocytoma, and GBM is diagnosed in the setting of IDH-wild-type status. It has been reported that IDH mutational status is an independent favorable prognostic factor for conferring longer progression-free and overall survival in GBM patients [7,8]. Moreover, patients with IDH-mutant grade 4 gliomas have been shown to exhibit a better prognosis than those with IDH-wild-type grade 3 gliomas. Collectively, these clinical findings emphasize the importance of determining IDH-mutant status in grade 4 astrocytomas [41]. The immunohistochemical assay is the most commonly used method for assessing IDH mutational status following invasive surgical interventions, which are associated with operative risks [42,43]. Moreover, the possibility of sampling error is highly relevant to determining histological grade and molecular profiling [11,44]. For example, IDH sequencing may be falsely negative if there are few glioma cells present within a tumor specimen [44] or substantial genetic heterogeneity occurs within the tumor specimen [11]. In addition, some exome sequencing studies have reported that traditional immunohistochemical assays do not detect IDH-mutant status in ~15% of gliomas [45]. Therefore, it is essential to develop non-invasive and objective imaging biomarkers for determining IDH mutational status in gliomas.

Mechanistically, wild-type IDH normally catalyzes the reversible, NADP+-dependent oxidative decarboxylation of isocitrate to alpha-ketoglutarate (α-KG) in the TCA cycle. However, IDH mutations confer a neomorphic enzyme activity converting α-KG to 2HG. Therefore, the oncometabolite 2HG has been proposed as a putative biomarker for IDH-specific genetic profiles for gliomas. A few studies have employed modified spectroscopic sequences and post-processing tools for detecting spectral resonances of 2HG from IDH-mutant gliomas [15,46,47,48]. However, the non-availability of these sequences and tools in the routine clinical setting renders these techniques less attractive. Moreover, diagnostic challenges may also arise due to the presence of a high degree of genetic heterogeneity within GBMs and partial sampling of these lesions, especially when single voxel spectroscopic methods are employed. In contrast, conventional MRI is a widely available, fast, easy-to-use, and economically affordable imaging modality that provides valuable information about brain tumor structural and morphological characteristics. Qualitative imaging features, such as frontal lobe tumor location, homogeneous signal intensity, sparse contrast enhancement within the tumor beds, and less intensive tumor infiltration are some of the imaging signatures that have been used to identify IDH-mutant gliomas with variable success [49,50,51]. However, all these qualitative associations were largely based on univariate analyses and hence, were prone to inter-observer variably. Therefore, a comprehensive analysis of imaging features is warranted for reliable prediction of IDH mutational status in spatially and temporally heterogeneous GBMs.

Radiomics is a quantitative analytical method of medical images that provides information that is generally difficult to perceive by visual inspection. Compared with conventional analytical approaches, radiomic analysis can provide a more efficient and unbiased quantification of imaging information. Readily interpretable and quantitative features, such as intensity distributions, spatial relationships, textural heterogeneity, and shape descriptors are extracted from a pre-defined ROI encompassing both solid and peritumoral regions of neoplasms in a typical fashion [52]. The training cohort is used to instruct the computer algorithm to detect patterns of features that are subsequently examined in a validation cohort to evaluate the algorithm’s performance in correctly predicting the presence or absence of a feature and its association with an outcome. In the recent past, the field of radiogenomics has been established to study the relationship between imaging features and underlying molecular processes and characteristics. Recently, it has been widely reported that radiomics/radiogenomics aids in guiding clinical decision making in neuro-oncology, particularly for making an accurate diagnosis, prognosis, and response assessment [23,24,25,26,27,53].

IDH mutation occurs only in 10% of grade 4 astrocytomas, so we could only include data from 23 IDH-mutant cases in the present study. Due to this small sample size and imbalance in data distribution, our data was prone to overfitting. Furthermore, in situations with an insufficient number of training datasets, the model is often overtrained. Consequently, the model performs well during the training stage but comparatively poorly during the subsequent testing stage. To address this challenge of small sample size, we leveraged the use of a well-established GAN method for synthesizing high-quality images and, in turn, raised the total sample size from 57 to 200. GAN is a deep learning architecture in which two neural networks compete against each other in a zero-sum game framework [54]. A GAN model consists of two components: a generator and a discriminator. In the training stage, the datasets produced by the generator, along with real images, serve as inputs to the discriminator. This can be considered comparable to enlarging the training datasets for the discriminator, whose purpose is to differentiate the real from the generated images [55]. Consequently, the discriminator will not immediately succumb to overfitting through the competitive relationship between these two networks, even when a limited number of training samples are used.

In a previous study [56], IDH mutational status was determined from a mixed population of grade III and grade IV gliomas. In the present study, only a histologically homogenous population of gliomas (grade IV astrocytomas) was included. Moreover, numerous radiomics features and machine learning classifiers were applied to predict IDH mutational status. Tumor necrosis was recognized as an important imaging feature and contributed most to the prediction model for distinguishing IDH-mutant grade 4 astrocytomas from IDH-wild-type GBMs when the K-best radiomics feature algorithm and decision tree (DT) classifier were used together. This finding is in agreement with an earlier study [56] in which IDH mutation was associated with a smaller enhancing volume and a larger necrotic volume when multiparametric radiomic profiles were analyzed. Additionally, imaging features from whole tumor volumes were found to be associated with IDH mutation status when the K-best radiomics feature selection algorithm and AB classifier were used together (AUC = 0.93). This finding may be explained by the fact that IDH-mutant gliomas have a more heterogeneous imaging microenvironment because of their stepwise gliomagenesis [57]. Our findings are also consistent with previous studies that have reported a larger tumor volume [58] and a lower degree of cellularity [59] in IDH-mutant compared to those in IDH-wild-type gliomas. Taken together, our results and published findings indicate that quantitative radiomic features can predict the IDH mutation status of grade 4 astrocytomas with high diagnostic power. However, these findings warrant further validation in multicentric, prospective studies with larger patient populations.

## 5. Conclusions

In conclusion, a prediction model based on conventional MRI-extracted radiomic features achieved promising diagnostic power in distinguishing IDH-mutant grade 4 astrocytomas from IDH-wild-type GBMs.

## Figures and Tables

**Figure 1 cancers-15-00951-f001:**
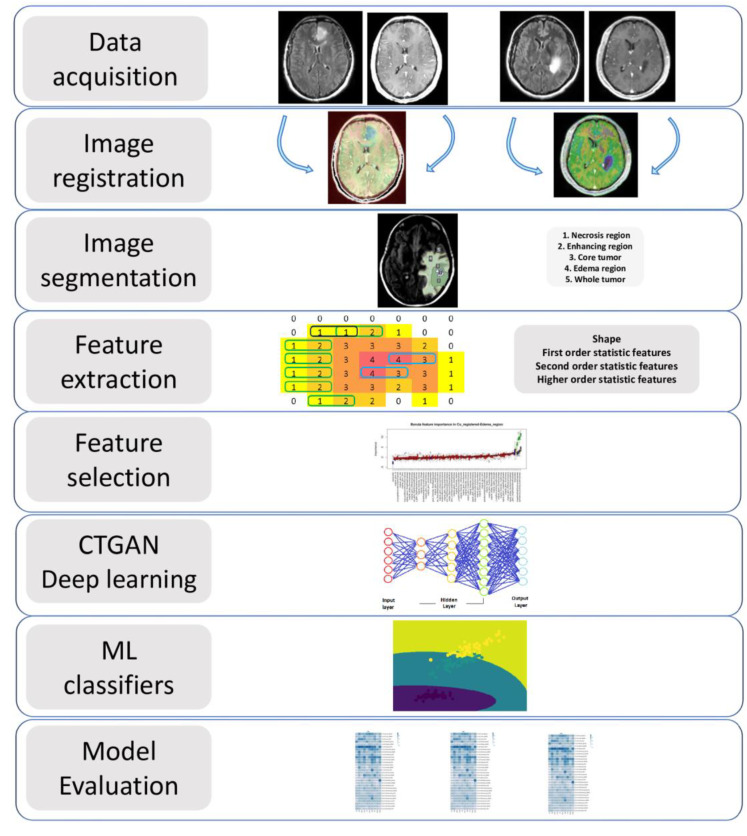
The overview of the image processing and radiomics pipeline.

**Figure 2 cancers-15-00951-f002:**
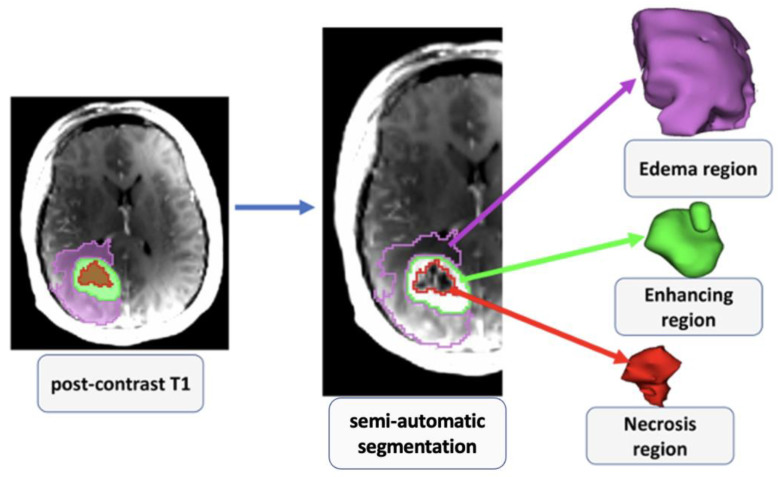
2D and 3D visualization of various subregions of a grade 4 astrocytoma as visible on post-contrast T1-weighted image.

**Figure 3 cancers-15-00951-f003:**
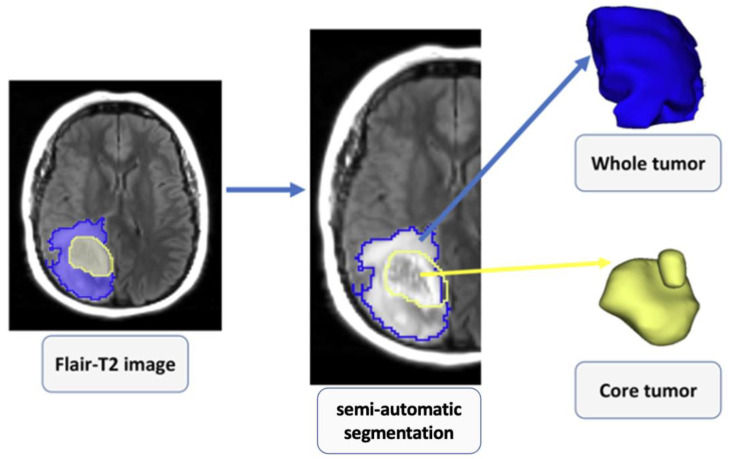
2D and 3D visualization of various subregions of a grade 4 astrocytoma as visible on T2-FLAIR image.

**Table 1 cancers-15-00951-t001:** Best 10 performances of multi-segmentation approaches, multi-machine learning classifiers, and multi-feature selection algorithms in discriminating IDH-mutant grade 4 astrocytomas from IDH-wild-type GBMs using original (Or) data set.

Radiomic Feature Combination	AUC	Accuracy	Sensitivity	Specificity	PR_AUC
Or_PC_T1_Core_AB_Kbest	0.93	0.92	1	0.86	0.92
Or_PC_T1_Core_KN_Kbest	0.93	0.92	1	0.86	0.92
Or_PC_T1_Core_LR_Kbest	0.93	0.92	1	0.86	0.92
Or_PC_T1_Core_MLP_Kbest	0.93	0.92	1	0.86	0.92
Or_T2-FLAIR_Enhancing_DT_Kbest	0.93	0.92	1	0.86	0.92
Or_T2-FLAIR_Enhancing_DT_mRmR	0.93	0.92	1	0.86	0.92
Or_T2-FLAIR_Enhancing_GB_mRmR	0.93	0.92	1	0.86	0.92
Or_T2-FLAIR_Enhancing_RF_mRmR	0.93	0.92	1	0.86	0.92
Or_PC_T1_Enhancing_HGB_RFE	0.93	0.92	1	0.86	0.92
Or_PC_T1_Enhancing_HGB_mRmR	0.93	0.92	1	0.86	0.92

**Table 2 cancers-15-00951-t002:** Best 10 performances of multi-segmentation approaches, multi-machine learning classifiers, and multi-feature selection algorithms in discriminating IDH-mutant grade 4 astrocytomas from IDH-wild-type GBMs using generated (Ge) data with 80% training set.

Radiomic Feature Combination	AUC	Accuracy	Sensitivity	Specificity	PR_AUC
Ge_PC_T1_Core_GNB_Kbest	0.93	0.92	1	0.86	0.92
Ge_PC_T1_Core_PA_RFE	0.93	0.92	1	0.86	0.92
Ge_T2_FLAIR_Enhancing_Bagging_Kbest	0.93	0.92	1	0.86	0.92
Ge_T2_FLAIR_Enhancing_DT_Kbest	0.93	0.92	1	0.86	0.92
Ge_T2_FLAIR_Whole_AB_Kbest	0.90	0.92	0.80	1	0.94
Ge_PC_T1_Core_RF_Kbest	0.90	0.92	0.80	1	0.94
Ge_PC_T1_Core_RF_RFE	0.90	0.92	0.80	1	0.94
Ge_PC_T1_Core_HGB_Kbest	0.90	0.92	0.80	1	0.94
Ge_PC_T1_Edema_AB_Kbest	0.90	0.92	0.80	1	0.94
Ge_PC_T1_Edema_Bagging_Kbest	0.90	0.92	0.80	1	0.94

**Table 3 cancers-15-00951-t003:** Best 10 performances of multi-segmentation approaches, multi-machine learning classifiers, and multi-feature selection algorithms in discriminating IDH-mutant grade 4 astrocytomas from IDH-wild-type GBMs using generated (Ge) data with 50% training set.

**Radiomic Feature Combination**	**AUC**	**Accuracy**	**Sensitivity**	**Specificity**	**PR_AUC**
Ge_PC_T1_Necrosis_Bagging_mRmR	0.92	0.92	0.91	0.94	0.93
Ge_PC_T1_Necrosis_RF_mRmR	0.92	0.92	0.91	0.94	0.93
Ge_PC_T1_Edema_KN_mRmR	0.92	0.92	0.91	0.94	0.93
Ge_PC_T1_Necrosis_KN_RFE	0.89	0.89	0.91	0.87	0.89
Ge_PC_T1_Edema_HGB_RFE	0.89	0.89	0.91	0.87	0.89
Ge_PC_T1_Necrosis_KN_RFE	0.88	0.89	0.82	0.94	0.90
Ge_PC_T1_Necrosis_KN_RFE	0.88	0.89	0.82	0.94	0.90
Ge_PC_T1_Edema_HGB_RFE	0.88	0.89	0.82	0.94	0.90
Ge_PC_T1_Edema_HGB_RFE	0.88	0.89	0.82	0.94	0.90
Ge_PC_T1_Core_KN_RFE	0.88	0.89	0.82	0.94	0.90

**Table 4 cancers-15-00951-t004:** Best 10 performances of multi-segmentation approaches, multi-machine learning classifiers, and multi-feature selection algorithms in discriminating IDH-mutant grade 4 astrocytomas from IDH-wild-type GBMs using generated (Ge) data with 30% training set.

Radiomic Feature Combination	AUC	Accuracy	Sensitivity	Specificity	PR_AUC
Ge_PC_T1_Core_LR_Kbest	0.91	0.92	0.86	0.96	0.92
Ge_PC_T1_Core_Ridge_Kbest	0.89	0.89	0.86	0.92	0.88
Ge_PC_T1_Core_SVC_mRmR	0.86	0.89	0.71	1	0.91
Ge_PC_T1_Core_LDA_Kbest	0.84	0.84	0.86	0.83	0.83
Ge_T2_FLAIR_Core_HGB_Kbest	0.82	0.79	0.93	0.71	0.80
Ge_T2_FLAIR_Core_LR_Kbest	0.81	0.84	0.71	0.92	0.83
Ge_PC_T1_Edema_GB_Kbest	0.81	0.84	0.71	0.92	0.83
Ge_T2_FLAIR_Core_LDA_Kbest	0.81	0.81	0.78	0.83	0.80
Ge_T2_FLAIR_Core_Ridge_Kbest	0.81	0.81	0.78	0.83	0.80
Ge_PC_T1_Enhancing_QDA_Kbest	0.80	0.79	0.86	0.75	0.79

## Data Availability

The data supporting this study’s findings and data processing algorithms will be available from the investigative team upon reasonable request.

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
