# Peer review of "MRI-Based Radiomics Combined with Deep Learning for Distinguishing IDH-Mutant WHO Grade 4 Astrocytomas from IDH-Wild-Type Glioblastomas"

_cancers, 2023, doi:10.3390/cancers15030951_

Round 1

Reviewer 1 Report

For authors,

Comment 1With this low number of patients, after splitting into test and training sets, how sure are you of the representability of the testing data set? Considering a small size, what is your justification of obtaining good AUC from ROC analysis? 

 Comment 2. What was the purpose of changing the percentage of split data into training and testing sets?

Comment 3. As there are a large number of methods available for extracting radiomic features, which method was used in this study and why.  How is your method different from previously published studies?

Reviewer 2 Report

- the study is well thought and displayed

- the cohort is not large considering the deep learning needs, but the patients are adequately enrolled (because naive IDHmutant grade 4 astrocytomas are unfrequent tumors)

- just two minors: 1) in the summary "whole tumor" is written twice (I guess one should be "core tumor"); 2) i suggest to erase the reduntant conclusion paragraph frome the Discussion and to live the identical phrase in the Conclusions.
